# Adult male-specific inverse association between dry eye disease and intraocular pressure: KNHANES 2010–2012

**Yun-Hee Choi**[1,2], **Martha Kim**[3], **Yoon-Hyeong Choi**[2,4]*, **Dong Hyun Kim**[1]*

**1** Department of Ophthalmology, Korea University College of Medicine, Seoul, South Korea, **2** School of Health and Environmental Science, Korea University, Seoul, South Korea, **3** Department of Ophthalmology, Dongguk University Ilsan Hospital, Goyang, Korea, **4** Institute of Health Science, Korea University, Seoul, South Korea

* amidfree@gmail.com (DHK); yoonchoi@korea.ac.kr (YHC)

## Abstract

### Purpose

To investigate the association between dry eye disease (DED) and intraocular pressure (IOP) in the general adult population of South Korea.

### Methods

We enrolled 13,194 adults (age $\geq$19 years) who had participated in the fifth Korea National Health and Nutrition Examination Survey conducted from 2010 to 2012. IOP was measured using Goldman applanation tonometry. DED was defined as the presence of self-reported symptoms along with a diagnosis by an ophthalmologist. As the correlation between the left and right eyes was high ($r$ = 0.833), only the measurement values of the right eye were used. We sequentially conducted multiple linear and logistic regression analyses to investigate the association between DED and IOP and prevalence of high IOP (>21 mmHg).

### Results

The prevalence of DED in Korean adults was 7.8%, and the geometric mean of IOP in the right eye was 13.7 ± 1.0 mmHg. In the fully adjusted model, participants with DED had a significantly lower IOP compared to those without DED in the overall population (β = -0.032; 95% confidence interval [CI]: -0.059, -0.004). This trend was evident in males (β = -0.059; 95% CI: -0.106, -0.012) but not in females. Similarly, males with DED had a significantly lower prevalence of high IOP compared to those without DED (odds ratio [OR] = 0.18; 95% CI: 0.04, 0.91), but this association was not evident in females.

### Conclusions

This study suggests that DED is associated with lower IOP in adult Korean males.

**Data Availability Statement:** The data underlying this study are owned by the Korean CDC and are available at the following link:

https://knhanes.kdca.go.kr/knhanes/main.do. All data files are available from the figshare database (https://doi.org/10.6084/m9.figshare.27079009).

**Funding:** This work was supported by Korea Environment Industry & Technology Institute (KEITI) through Core Technology Development Project for Environmental Diseases Prevention and Management funded by Korea Ministry of Environment (MOE) (grant number: RS-2022-KE002107 to DHK).

**Competing interests:** None of the authors has any conflicts of interest to disclose.

**Abbreviations:** BMI, body mass index; CB1, cannabinoid receptor 1; CCT, central corneal thickness; CI, confidence interval; DED, dry eye disease; GM, geometric mean; IOP, intraocular pressure; KNHANES, Korea National Health and Nutrition Examination Survey; OR, odds ratio; PG, prostaglandin; TBUT, tear break-up time.

## Introduction

Dry eye disease (DED) is characterized by inadequate tear production and the presence of an unstable tear film on the ocular surface, leading to sensations of irritation, discomfort, and compromised visual quality [1, 2]. Its prevalence has been steadily increasing over the past 30 years because of air pollution, exposure to chemicals, and the use of digital devices [3]. In 2021, 11.6% of the world's population was estimated to suffer from DED [4]. In Korea, the prevalence of DED ranges from 16.2% to 33.2% and has been increasing by an average of 2.1% per year over the past 5 years [5–7]. The treatment and management of DED have become a major public health concern.

Intraocular pressure (IOP), the fluid pressure in eyes, is maintained by the balance of aqueous humor secretion and drainage [8, 9]. Dysregulation of IOP can deform the globe and cause eye disorders. Low IOP (<5 mmHg) may lead to globe contraction, choroidal detachment, and hypotony maculopathy [10], while high IOP (>21 mmHg) can damage the optic nerve and retinal ganglion cells, causing glaucomatous visual field defects [11, 12].

IOP is related to lifestyle and health conditions [13, 14]. It is usually higher in males [13], and increases with age [15], while smoking, alcohol consumption, obesity, diabetes, and hypertension also contribute as risk factors for elevated IOP [13–17]. Additionally, certain ocular conditions (e.g., thyroid eye disease) can affect IOP, potentially causing complications like glaucoma. Therefore, identifying and minimizing these factors are crucial [18, 19].

Among ocular conditions that affect IOP, DED can influence the outflow of aqueous humor due to inflammation on ocular surface [20]; however, the association between DED and IOP remains unclear. Some animal studies showed that the expression of certain prostaglandins (PGs) such as PGF2α can lower IOP in dry eyes [20, 21], but only few human studies have been conducted and have reported inconsistent findings. One study found a significant inverse correlation between IOP and corneal staining score, which showed higher values in DED patients, regardless of their glaucoma status [22]. Another study showed that IOP was significantly lower in patients with a short tear break-up time (TBUT), a characteristic of DED [23]. However, a study on DED patients reported that while IOP was lower in these patients, the relationship didn't reach statistical significance [24]. Therefore, more evidence is required to confirm the association between DED and IOP in large human samples.

Therefore, the present study aimed to investigate the association between DED and IOP in Korean adults using data from the fifth Korea National Health and Examination Survey (KNHANES) conducted from 2010 to 2012. In addition, considering the influence of sex-specific differences in IOP [25], we aimed to assess whether this association differed between the sexes.

## Materials and methods

### Study participants

The KNHANES is a nationwide cross-sectional survey conducted by the Korea Centers for Disease Control and Prevention Agency (KCDC) since 1998 to assess the health and nutritional status of the general population of South Korea. It comprised three main components: health survey, medical examination, and nutrition survey. The data from this survey are publicly available.

We enrolled participants aged ≥19 years who had participated in the fifth wave of KNHANES (conducted from 2010 to 2012). As the KNHANES provided information on DED exclusively during 2010–2012, we used the data from this period. Among the 25,534 individuals initially recruited in this wave, 19,599 were aged ≥19 years. After excluding 2,880 adults

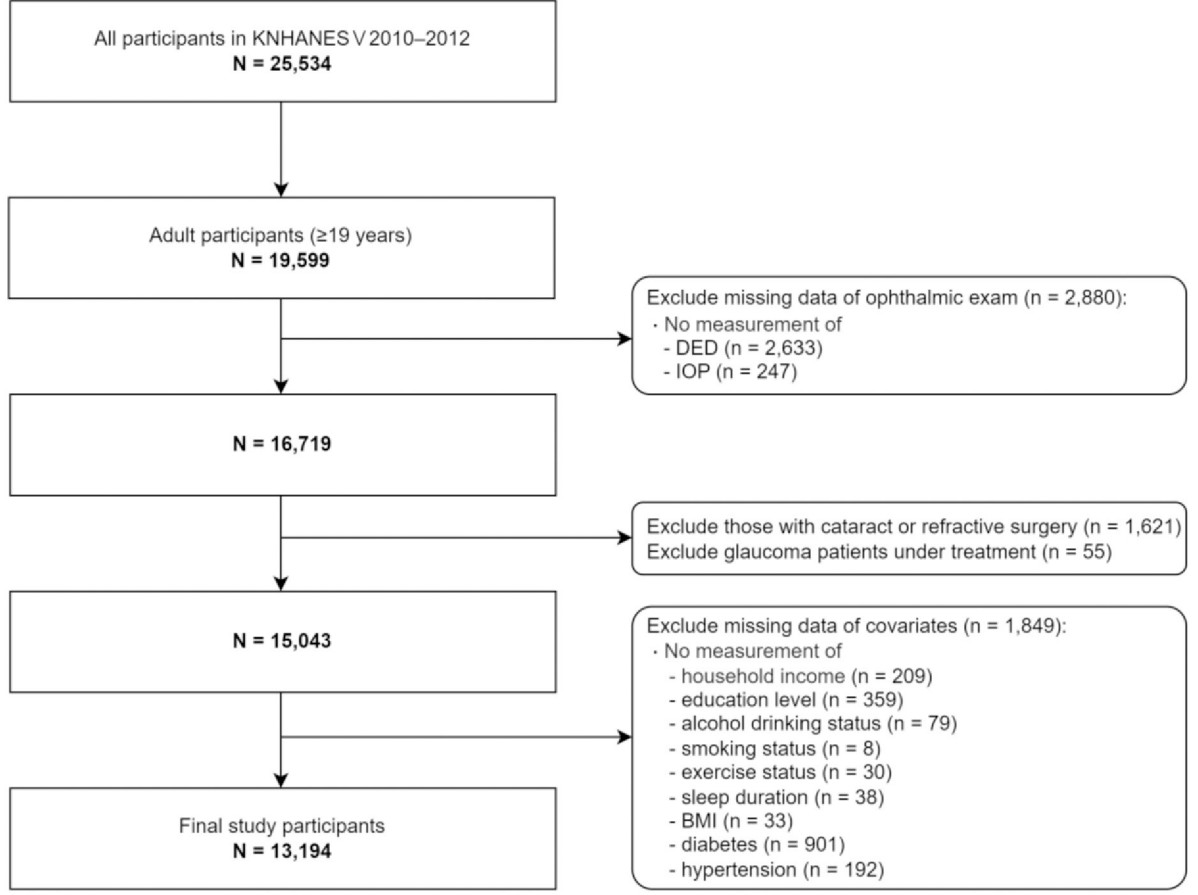

**Fig 1. Flowchart of exclusion criteria and final number of study participants in Korean National Health and Examination Survey conducted from 2010 to 2012.** Abbreviations: BMI, body mass index; DED, dry eye disease; IOP, intraocular pressure.

who had not undergone ocular examinations for DED or IOP, we sequentially excluded participants with previous cataract or refractive surgery (n = 1,621) and patients with glaucoma who were under treatment (n = 55). Furthermore, we excluded 1,849 adults with no information on the following covariates: household income (n = 209), education level (n = 359), alcohol drinking status (n = 79), smoking status (n = 8), exercise status (n = 30), sleep duration (n = 38), body mass index (BMI; n = 33), diabetes (n = 901), or hypertension (n = 192). Finally, 13,194 individuals were included in the study (Fig 1).

In this study, 13,194 participants represented the Korean population, which was estimated at 16,600,400 individuals (weighted). The study protocol was approved by the Institutional Review Board of the KCDC (approval number: 2010–02CON-21-C in 2010, 2011–02CON-06-C in 2011, and 2012–01EXP-01–2C in 2012), and all participants provided written informed consent.

## Definition of DED

Assessment of DED was conducted using a self-reported questionnaire among aged 19 years and older. The DED questionnaire items were constructed based on a standardized guideline developed by the Korean Ophthalmological Society for the assessment of ocular health among

the Korean population [26]. The following two questionnaire items were used to assess DED: "Have you ever been diagnosed with DED by a medical doctor?" and "Have you ever experienced dry eye symptoms such as dryness or irritation in your eyes, excluding cases where symptoms occurred occasionally or intermittently?". Participants responded each question with a "yes" or "no". Those who responded "yes" in both diagnosis and symptom were categorized under the DED group [27].

## Measurement of IOP and detection of a high IOP

Ophthalmologists affiliated with the National Epidemiology Survey Committee of the Korean Ophthalmological Society conducted the ophthalmic examinations. After conducting a health survey that included a medical history related to ophthalmic diseases, IOP was measured once in both the left and right eyes using a Goldmann applanation tonometer (Haag-Streit; Haag-Streit AG, Koeniz, Switzerland). High IOP was defined as an IOP >21 mmHg [28].

## Covariates

Covariates were selected based on previous epidemiological studies and biological mechanisms. We considered the following variables as potential covariates: age, sex, survey year, region, household income, education level, alcohol drinking, smoking, exercise, sleep duration, BMI, and a family history of glaucoma, diabetes, and hypertension [13, 17, 29–32]. The correlation among the covariates was low (Pearson's correlation coefficient $r < 0.5$, $p < 0.05$), indicating that the influence of multicollinearity could be excluded. Age and BMI were fitted as continuous variables. Sex (male or female), survey year (2010, 2011, or 2012), region (urban or rural), household income (low, low to medium, medium to high, or high), education level (below high school, high school, college or higher), alcohol consumption (never, former, or current), smoking status (never, former, or current), exercise status (little, moderate, or vigorous), sleep duration (<7 or ≥7 h), and a family history of glaucoma (no or yes), diabetes (no or yes), and hypertension (no or yes) were considered as categorical variables. Regions were categorized as urban (Seoul, Gyeonggi, Busan, Daegu, Incheon, Gwangju, Daejeon, and Ulsan) or rural (Gangwon, Chungbuk, Chungnam, Jeonnam, Jeonbuk, Gyeongbuk, Gyeongnam, and Jeju). Household income was classified according to the income criteria defined by Statistics Korea. BMI was calculated by dividing the weight (kg) by height squared (m²). Diabetes was determined based on a fasting blood glucose level ≥126 mg/dL, self-reported current diabetic medication or insulin injections, or a physician's diagnosis. Hypertension was determined based on a systolic blood pressure ≥140 mmHg or a diastolic blood pressure ≥90 mmHg, self-reported current antihypertensive medication, or a physician's diagnosis.

## Statistical analyses

The KNHANES was designed to represent the general population of Korea using a stratified multistage probability sampling method [33]. Therefore, in this study, we integrated data from the KNHANES (2010–2012) and conducted statistical analyses by applying weights to the participants according to this method.

The IOP showed a right-skewed distribution; therefore, it was transformed using a logarithmic transformation to achieve a normal distribution. Student's *t*-test and Wald's F-test were used to compare the geometric mean (GM) of IOP according to the demographic characteristics of the participants; whereas, the Rao-Scott chi-square test was used to compare the prevalence of DED.

A multiple linear regression analysis was performed to investigate the association between DED and IOP. DED was categorized as a binary variable (presence or absence), while IOP was

treated as a continuous variable. The association was assessed by comparing the effects of DED on IOP level and estimating regression coefficients (β) and 95% confidence intervals (CIs). Additionally, a logistic regression analysis was performed to assess the effect of DED on the prevalence of high IOP. DED and high IOP were categorized as a binary variable. The association was assessed by comparing individuals without DED to the reference group and estimating the odds ratios (ORs) and 95% CIs. We developed three sequential models, differentiated by (1) demographic variables, (2) anthropometric measurements and health-related behaviors, and (3) genetic or disease variables, to assess their influence on the outcome [34]. Model 1 was adjusted for age, sex, survey year, region, household income, and education level. Model 2 was adjusted for all variables included in model 1 and further for alcohol drinking, smoking, exercise, sleep duration, and BMI. Model 3 was adjusted for all variables included in model 2 and further for a family history of glaucoma, diabetes, and hypertension.

We also conducted fully adjusted associations between DED and IOP in subgroups stratified by age (20–39 or ≥40 years), survey year (2010, 2011, or 2012), region (urban or rural), household income (low, low to medium, medium to high, or high), education level (below high school, high school, or college or higher), alcohol consumption (never, former, or current), smoking status (never, former, or current), exercise status (vigorous, moderate, or little), sleep duration ($<7$ or $\geq 7$ h), BMI ($<25$ or $\geq 25$ kg/m$^2$), family history of glaucoma (no or yes), diabetes (no or yes), and hypertension (no or yes). To evaluate the potential interaction effects of each subgroup variable as a latent effect modifier, we added a product term for each stratified individual variable and DED status to the regression model and assessed them using likelihood ratio tests.

Finally, previous studies have reported that there are sex-specific differences in IOP [13]. Therefore, in this study, all analyses were stratified by sex to examine the association between DED and IOP.

All statistical analyses were performed using the "survey" package in R (version 4.3.1.; R Development Core Team). Multiple imputations were analyzed using the "mice" package. A $p$-value $< 0.05$ with a two-sided test was set to determine statistical significance.

## Sensitivity analyses

Sensitivity analyses were conducted to confirm the significance of the results of the various models. First, we evaluated whether or not the results are consistent even with IOP in the left eye. Second, we analyzed the models without applying complex sampling weights. Third, for covariates with missing values (14.01% of the cases), multiple imputations were conducted by adding values using multiple imputation methods. Ten sets of input data were generated to estimate the variance in the input values. The estimates from multiple linear or logistic regression analyses for each dataset were then integrated to obtain the final estimates [35].

## Results

### Demographic characteristics of study participants

Table 1 presents the demographic characteristics of the study participants according to the DED and IOP distribution. Of the total participants, 1,034 (7.8%) had DED. The prevalence of DED was significantly higher in females than in males ($p < 0.05$). The weighted GM (95% CI) for IOP among all participants was 13.7 (13.6, 13.9) mmHg. IOP was significantly lower in adults aged 20–39 years compared to those aged ≥40 years ($p < 0.001$) and in females compared to males ($p < 0.001$). Furthermore, it was significantly lower among participants living in urban regions, non-smokers, those who slept for ≥7 h, those who had a BMI $<25$ kg/m$^2$, and those without diabetes or hypertension (all $p < 0.05$).

**Table 1. Demographic and clinical characteristics of study participants according to the dry eye disease (DED) and the intraocular pressure (IOP) (n = 13,194).**

| Characteristics | DED | | p-value[a] | IOP (mmHg) | p-value |
|---|---|---|---|---|---|
| | Case, n (%) | Control, n (%) | | GM (95% CI) | |
| Total | 1,034 (7.8) | 12,610 (92.2) | | 13.7 (13.6, 13.9) | |
| Age (years) | | | 0.286 | | <0.001 |
| 20–39 | 286 (7.1) | 3,744 (92.9) | | 13.7 (13.5, 13.8) | |
| ≥40 | 748 (8.2) | 8,416 (91.8) | | 13.8 (13.7, 13.9) | |
| Sex | | | <0.001 | | <0.001 |
| Male | 223 (3.9) | 5,478 (96.1) | | 13.9 (13.8, 14.0) | |
| Female | 811 (10.8) | 6,682 (89.2) | | 13.6 (13.5, 13.7) | |
| Survey year | | | <0.001 | | 0.914 |
| 2010 | 310 (7.1) | 4,063 (92.9) | | 13.8 (13.6, 14.0) | |
| 2011 | 288 (6.3) | 4,310 (93.7) | | 13.8 (13.6, 14.0) | |
| 2012 | 436 (10.3) | 3,787 (89.7) | | 13.7 (13.5, 13.9) | |
| Region | | | 0.041 | | <0.001 |
| Urban | 759 (8.6) | 8,110 (91.4) | | 13.7 (13.6, 13.9) | |
| Rural | 275 (6.4) | 4,050 (93.6) | | 13.9 (13.7, 14.1) | |
| Household income | | | 0.609 | | 0.916 |
| Low | 155 (7.0) | 2,068 (93.0) | | 13.7 (13.5, 13.9) | |
| Low to medium | 262 (7.0) | 3,459 (93.0) | | 13.7 (13.6, 13.9) | |
| Medium to high | 301 (8.0) | 3,479 (92.0) | | 13.7 (13.6, 13.9) | |
| High | 316 (8.5) | 3,416 (91.5) | | 13.8 (13.7, 14.0) | |
| Education level | | | 0.764 | | 0.362 |
| Below high school | 354 (7.8) | 4,206 (92.2) | | 13.7 (13.5, 13.8) | |
| High school | 380 (8.2) | 4,238 (91.8) | | 13.8 (13.6, 13.9) | |
| College or higher | 300 (7.5) | 3,716 (92.5) | | 13.8 (13.6, 13.9) | |
| Alcohol drinking status | | | 0.003 | | 0.020 |
| Never | 167 (9.8) | 1,534 (90.2) | | 13.7 (13.5, 13.9) | |
| Former | 161 (8.9) | 1,658 (91.1) | | 13.5 (13.4, 13.7) | |
| Current | 706 (7.3) | 8,968 (92.7) | | 13.8 (13.7, 13.9) | |
| Smoking status | | | <0.001 | | 0.001 |
| Never | 782 (10.0) | 7,011 (90.0) | | 13.6 (13.5, 13.8) | |
| Former | 147 (5.6) | 2,486 (94.4) | | 13.9 (13.7, 14.0) | |
| Current | 105 (3.8) | 2,663 (96.2) | | 13.9 (13.7, 14.0) | |
| Exercise status | | | 0.119 | | 0.476 |
| Little | 851 (8.0) | 9,841 (92.0) | | 13.8 (13.6, 13.9) | |
| Moderate | 43 (6.0) | 670 (94.0) | | 13.6 (13.2, 13.9) | |
| Vigorous | 140 (7.8) | 1,649 (92.2) | | 13.8 (13.6, 13.9) | |
| Sleep duration (h) | | | <0.001 | | 0.014 |
| <7 | 483 (9.0) | 4,905 (91.0) | | 13.8 (13.7, 13.9) | |
| ≥7 | 551 (7.1) | 7,255 (92.9) | | 13.6 (13.5, 13.8) | |
| BMI (kg/m$^2$) | | | 0.042 | | <0.001 |
| <25 | 732 (8.2) | 8,204 (91.8) | | 13.6 (13.5, 13.7) | |
| ≥25 | 302 (7.1) | 3,956 (92.9) | | 14.1 (13.9, 14.2) | |
| Family history of glaucoma | | | 0.307 | | 0.909 |
| No | 999 (7.8) | 11,888 (92.2) | | 13.8 (13.7, 14.0) | |
| Yes | 35 (11.4) | 272 (88.6) | | 13.8 (13.4, 14.2) | |
| Diabetes | | | 0.089 | | <0.001 |
| No | 950 (7.9) | 11,002 (92.1) | | 13.7 (13.6, 13.8) | |

*(Continued)*

**Table 1.** (Continued)

| Characteristics | DED | | p-value[a] | IOP (mmHg) | p-value |
|---|---|---|---|---|---|
| | Case, n (%) | Control, n (%) | | GM (95% CI) | |
| Yes | 84 (6.8) | 1,158 (93.2) | | 14.2 (14.0, 14.5) | |
| Hypertension | | | 0.082 | | <0.001 |
| No | 739 (8.1) | 8,397 (91.9) | | 13.7 (13.5, 13.8) | |
| Yes | 295 (7.3) | 3,768 (92.7) | | 14.0 (13.8, 14.1) | |

[a]Student's *t*-test, Wald F-test, or Rao-Scott chi-square test

Abbreviations: BMI, body mass index; CI, confidence interval; DED, dry eye disease; IOP, intraocular pressure; GM, geometric mean

Finally, owing to the strong positive correlation between IOP in the left and right eyes ($r$ = 0.833; S1 Fig), only data from the right eye are presented.

## Association between DED and IOP

Table 2 presents the association between DED and IOP according to the covariate-adjusted models. In the fully adjusted model (model 3), participants with DED in the overall population had a significantly lower IOP compared to those without DED (β = -0.032; 95% CI: -0.059, -0.004). When stratified by sex, males with DED showed a significantly lower IOP compared to males without DED (β = -0.059; 95% CI: -0.106, -0.012). In females, IOP tended to decrease with DED, although without statistical significance.

Table 3 presents the associations between DED and the prevalence of high IOP (>21 mmHg) according to the covariate-adjusted models. In the fully adjusted model (model 3), participants with DED in the overall population had a marginally significantly lower prevalence of high IOP compared to those without DED (OR = 0.43; 95% CI: 0.17, 1.06). When stratified by sex, males with DED showed a significantly lower prevalence of high IOP compared to males without DED (OR = 0.18; 95% CI: 0.04, 0.91). The prevalence of high IOP tended to be lower in females in the presence of DED, although this trend lacked statistical significance.

**Table 2. Multiple linear regression analysis results for the effects of DED on IOP (n = 13,194).**

| Variables | Total | Male | Female |
|---|---|---|---|
| | β (95% CI) | β (95% CI) | β (95% CI) |
| Model 1 | | | |
| DED vs. no DED | -0.005 (-0.021, 0.011) | -0.023 (-0.051, 0.005) | 0.001 (-0.019, 0.021) |
| Model 2 | | | |
| DED vs. no DED | -0.003 (-0.019, 0.012) | *-0.024 (-0.052, 0.005)* | 0.003 (-0.017, 0.023) |
| Model 3 | | | |
| DED vs. no DED | **-0.032 (-0.059, -0.004)** | **-0.059 (-0.106, -0.012)** | -0.020 (-0.055, 0.015) |

**Bold**: $p < 0.05$, *Italic*: $p < 0.1$

Model 1: adjustment for age, sex, survey year, region, income, and education

Model 2: model 1 + adjustment for alcohol drinking status, smoking status, exercise status, sleep duration, and body mass index

Model 3: model 2 + adjustment for family history of glaucoma, diabetes, and hypertension

Abbreviations: CI, confidence interval; DED, dry eye disease

**Table 3. Multiple logistic regression analysis results for the effects of DED on high IOP (>21 mmHg) (n = 13,194).**

| Variables | Total | Male | Female |
|---|---|---|---|
| | OR (95% CI) | OR (95% CI) | OR (95% CI) |
| Model 1 | | | |
| DED vs. no DED | **0.40 (0.16, 0.99)** | **0.18 (0.04, 0.88)** | 0.53 (0.18, 1.56) |
| Model 2 | | | |
| DED vs. no DED | *0.42 (0.17, 1.04)* | **0.18 (0.04, 0.89)** | 0.55 (0.19, 1.59) |
| Model 3 | | | |
| DED vs. no DED | *0.43 (0.17, 1.06)* | **0.18 (0.04, 0.91)** | 0.57 (0.20, 1.65) |

**Bold**: $p < 0.05$; *Italic*: $p < 0.1$

Model 1: adjustment for age, sex, survey year, region, income, and education

Model 2: model 1 + adjustment for alcohol drinking status, smoking status, exercise status, sleep duration, and body mass index

Model 3: model 2 + adjustment for family history of glaucoma, diabetes, and hypertension

Abbreviations: CI, confidence interval; DED, dry eye disease; OR, odds ratio

## Subgroup analysis on the association between DED and IOP

Fig 2 presents the results of the subgroup analysis of the association between DED and IOP in the overall population and separately for males and females. In the overall population, participants residing in urban areas exhibited a significantly stronger inverse association between DED and IOP compared to their rural counterparts (urban: -0.051 [-0.086, -0.016] vs. rural: 0.026 [-0.019, 0.072]; $p$-interaction $< 0.05$). Among males, the inverse association was significantly stronger in participants with more than medium level of household income (high: -0.076 [-0.144, -0.007] vs medium to high: -0.175 [-0.262, -0.087] vs low to medium: 0.001 [-0.098, 0.100] vs low: 0.042 [-0.042, 0.126]), college or higher level of education (college or higher: -0.101 [-0.164, -0.038] vs high school: -0.032 [-0.213, 0.148] vs below high school: -0.030 [-0.104, 0.044]), and those without hypertension (no: -0.083 [-0.136, -0.031] vs. yes: 0.003 [-0.090, 0.096]) (all $p$-interaction $< 0.05$). In contrast, female participants showed no significant differences between the groups.

According to the sensitivity analysis results of accounting for the left eye, DED and IOP showed inverse associations in the overall population along with males in the fully adjusted model (model 3; S1 and S2 Tables). Even in models without complex sample weights, males with DED showed a significant inverse association with IOP in the fully adjusted model (model 3; S3 and S4 Tables). In the analysis in which the missing values had been imputed using multiple imputation methods for participants without covariates, significant results observed in the main analysis persisted in the fully adjusted model (model 3; S5 and S6 Tables).

## Discussion

This study utilized data from the KNHANES (2010–2012), which represents the general population of South Korea, and identified that the prevalence of DED increased from 7.1% to 10.3% during this period. This figure is similar to the 8% to 10.4% prevalence reported in national statistics for the same period [36, 37], which is likely influenced by air pollution levels and the widespread use of smartphones [36, 37]. Moreover, in our main findings, we found a significant negative association between DED and IOP among males. However, female participants

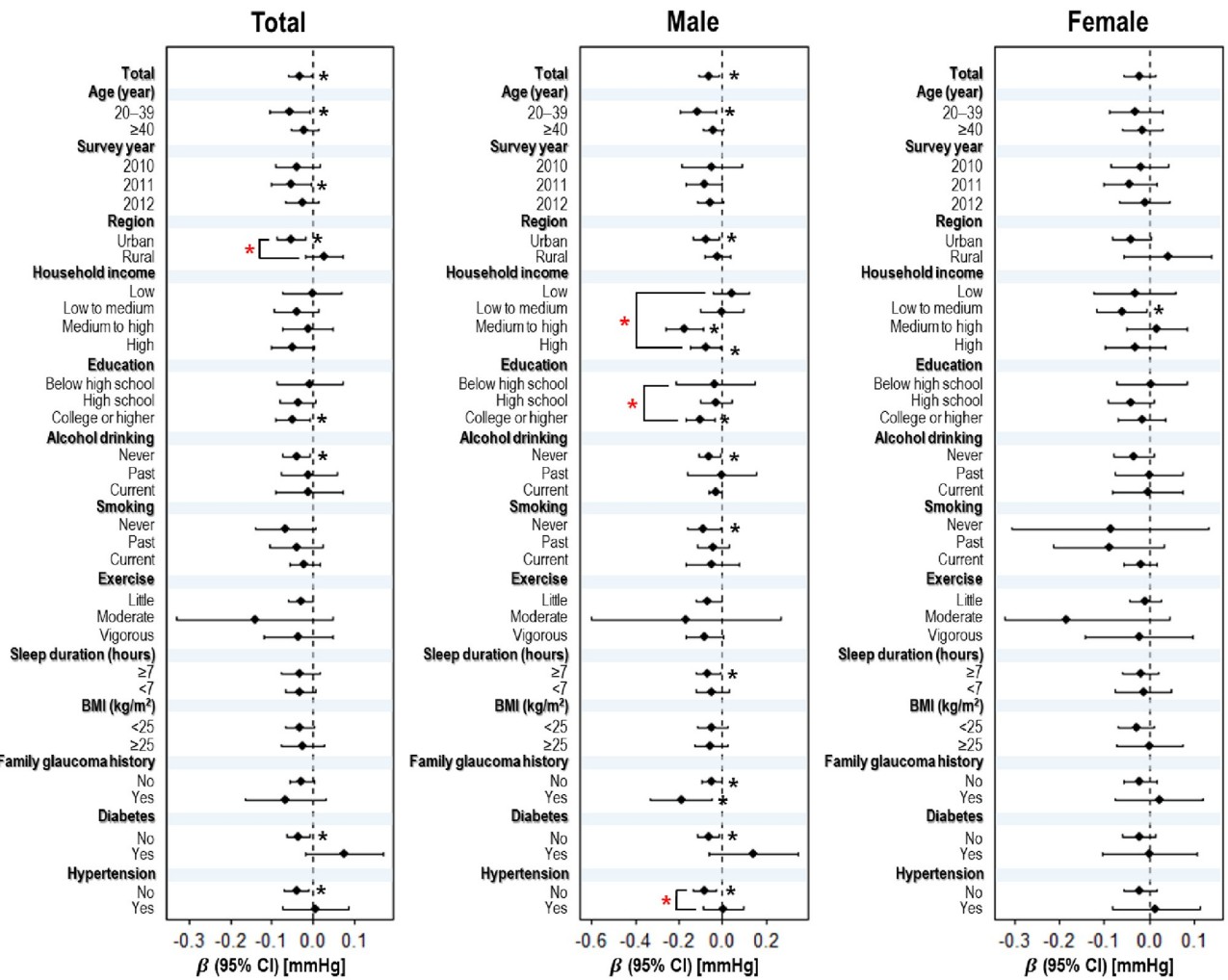

**Fig 2. Association between DED and IOP among subgroups.** Black asterisks (*) indicate significant associations in each group ($p < 0.05$). Red asterisks indicate significant differences between the groups ($p$-interaction $< 0.05$). Abbreviations: BMI, body mass index; DED, dry eye disease; IOP, intraocular pressure; CI, confidence interval.

showed no significant association. Furthermore, association was more pronounced among males with higher household income or education levels and without hypertension.

To our knowledge, this is the first epidemiological study to investigate the association between DED and IOP. Therefore, a direct comparison of the results of this study with those of previous studies is challenging. A study on Japanese glaucoma patients reported that both in glaucoma and non-glaucoma patients, those with DED had lower IOP compared to patients without DED. The study also showed that IOP was negatively correlated with corneal staining score [22]. Another study on Japanese found that IOP was lower in those with a short TBUT [23]. Although the direct association between DED and IOP has not been confirmed, some clinical studies have reported the expression of specific PGs that can affect the lowering of IOP in DED patients. A clinical study involving 40 individuals reported a positive association between the PGF2α expression levels and increased corneal staining [38]. Another clinical study involving 36 individuals reported a positive association between the PGE2 levels in tears and the severity of DED symptoms [39]. PGE2 or PGF2α reduced IOP by increasing the

outflow of aqueous humor on the ocular surface as a substance mediating inflammatory responses [29, 40]. Therefore, although these studies involved small sample sizes of <50 individuals, they support the findings of this study that DED is inversely associated with IOP.

The results of this study can be explained by biological mechanisms elucidated in previous experimental studies. In a mouse study, the PGE2 or PGF2α concentration in tears significantly increased among patients with DED compared to those without DED [41]. In another mouse experiment, there was increased expression of PGF2α synthase and elevated PGF2α levels on the ocular surface of the dry eye model [42]. Another *in vitro* study verified that hyperosmolarity, a key stress in dry eye upregulated cyclooxygenase-2 and PGF2α release from human corneal epithelial cells [43]. Specialized pro-resolving mediators derived from polyunsaturated fatty acids, which act as resolution mediators for immune responses on the ocular surface, can also induce inflammatory reactions and trigger PG secretion, especially PGF2α, in response to damage caused by DED [44]. Moreover, sensation of DED can be associated with elevated systemic inflammatory markers (e.g., interleukin-6, tumor necrosis factor-alpha) and pain modulation in central nervous system [45, 46]. A representative mechanism is the increase in uveoscleral outflow of aqueous humor triggered by $PGF_{2\alpha}$, thereby lowering IOP [47]. Drugs modified based on these mechanisms are used to lower IOP and treat glaucoma [48]. Furthermore, systematic inflammation can influence the regulation of IOP outflow, and when it inhibits aqueous humor production, it may lead to a reduction in IOP [49]. Therefore, the results of this study may be related with PG-mediated response of DED.

A notable finding from this study is that the inverse association between DED and IOP was more pronounced in males than in females. Epidemiological studies investigating sex-specific effects on DED and IOP changes are scarce, with only a few clinical studies suggesting that the association between DED and IOP varies according to sex. A study in elderly individuals reported that DED in females decreased corneal epithelial cell density, damaged corneal nerve cells, and subsequently influenced aqueous outflow [50]. Studies focusing on postmenopausal women reported differences in tear secretion depending on the influence of sex hormones in postmenopausal women with DED [51, 52]. Although these studies targeted specific populations, they indicated that there are sex-dependent associations between DED and IOP.

The sex-specific mechanisms underlying the effects observed here can be explained by a previous study, suggesting that responses to DED and IOP regulation may differ according to sex of the individual. First, androgens are known to act as anti-inflammatory agents in the inflammatory response occurring in dry eyes. Previous experimental studies have suggested that androgens downregulate the expression of inflammatory response genes in the ocular surface epithelial cells of dry eye patients [53]. In another study, lipid mediators such as PGs in human tears showed higher expression in men in response to immune reactions caused by dry eye, while the levels of specialized pro-resolving mediators and PGs were significantly lower in women's tears [54]. Furthermore, cannabinoids, which function as neurotransmitters and consist of receptors (CB1, CB2, and others), are upregulated by corneal damage and contribute to healing, resulting in decreased IOP [55, 56]. Several animal studies have reported that male mice exhibit higher levels of CB1 receptor mRNA compared to female mice, suggesting that the IOP-lowering effect induced by CB1 is more pronounced in males [56, 57]. Therefore, these sex-specific effects observed during the healing process of DED support the greater association between DED and lower IOP in males than in females.

As evidenced by multiple covariate adjustment models and subgroup analyses, the effect of DED on IOP varied according to demographic and disease-related characteristics. The inverse association between DED and IOP was not significant in models 1 and 2 (Table 2), which adjusted for demographic characteristics and health-related behaviors. However, a significant association was observed in model 3, which additionally adjusted for genetic and disease-

related factors, suggesting that family history of glaucoma, diabetes, and hypertension play crucial roles in this relationship. This finding was further elucidated in subgroup analyses, which revealed a more pronounced inverse association between DED and IOP among males with higher household income and education levels who did not have hypertension. Low household income and education level were well-established risk factors for glaucoma [58]. Additionally, increase in systolic blood pressure and heart rate influences in rise in IOP [59]. Although the differences between groups were not significant, a negative association between DED and IOP was observed among males who were young adults (20–39 years), urban residents, non-drinkers, non-smokers, slept more than 7 hours, and those without diabetes. It is known that age over 40 years, alcohol consumption, smoking, sleep deprivation, and diabetes are considered risk factors for elevated IOP and glaucoma. These factors contribute to the risk through various mechanisms, including osmotic effects [17], vascular problems [16, 60], and alterations in sleep hormones [30] or aqueous outflow [61]. Therefore, the results of this study indicate that different treatment strategies for glaucoma management should be considered for males with lower household income, education levels, or hypertension, along with consideration of their lifestyle factors.

This study had several limitations. First, the KNHANES used in this study had a cross-sectional survey design; therefore, causal relationships between DED and IOP cannot be guaranteed. However, the results of some animal experiments describing the molecular and biological mechanisms underlying IOP reduction in patients with DED support the findings of the present study [21, 53]. Second, DED was ascertained based on both self-reported symptoms and physician diagnoses. Therefore, DED prevalence reported in this study may differ from the actual prevalence in the Korean population [36, 37]. Moreover, patients' willingness to seek medical attention, despite experiencing similar symptoms, may lead to misclassification of DED. However, since we employed a more conservative indicator compared to studies that rely solely on symptoms, actual effect may be more pronounced. Third, the KNHANES data did not provide information on meibomian gland dysfunction; therefore, age-related changes in the meibomian glands could not be considered. Fourth, information on hormone therapy or medication use was unavailable in the KNHANES and not included in the analysis. Fifth, KNHANES used a Goldmann applanation tonometer, a gold standard for IOP measurement. This technique, while more robust than noncontact pneumotonometry, remains partially influenced by corneal hysteresis and sensation [24]. The IOP measurements in this study were consistently obtained from the right eyes, while the DED status was not differentiated between eyes. Consequently, while we acknowledge the potential for measurement errors inherent to this technique, we cannot definitively ascertain whether these potential inaccuracies significantly impacted our findings. Further clinical studies on DED patients are necessary to confirm this effect. Despite these limitations, we utilized KNHANES data representing the Korean adult population, allowing for the interpretation and generalization of the observed results to the Korean population. Additionally, by adjusting for various covariates related to demographic characteristics, lifestyle factors, and health status, the results can be interpreted while excluding potential confounding factors. Therefore, our findings provide reliable epidemiological evidence that DED is associated with lower IOP.

## Conclusions

This study demonstrates a significant negative association between DED and IOP in Korean males. Our study is expected to be a pioneering epidemiological study involving a large-scale general population, observing the relationship between DED and IOP for the first time.

Further longitudinal studies targeting different populations are required to validate the findings of this study.

## Supporting information

**S1 Fig. Scatter plot showing the correlation between IOP in the left and right eyes.** IOP, intraocular pressure.
(DOCX)

**S1 Table. Multiple linear regression analysis results for the effects of DED on IOP in the left eye (n = 13,194).**
(DOCX)

**S2 Table. Multiple logistic regression analysis results for the effects of DED on high IOP (>21 mmHg) in the left eye (n = 13,194).**
(DOCX)

**S3 Table. Multiple linear regression analysis results for the effects of DED on IOP after excluding sampling weights (n = 13,194).**
(DOCX)

**S4 Table. Multiple logistic regression analysis results for the effects of DED on high IOP (>21 mmHg) after excluding sampling weights (n = 13,194).**
(DOCX)

**S5 Table. Multiple linear regression analysis results for the effects of DED on IOP after incorporating multiple imputations (n = 15,043).**
(DOCX)

**S6 Table. Multiple logistic regression analysis results for the effects of DED on high IOP (>21 mmHg) after incorporating multiple imputations (n = 15,043).**
(DOCX)

## Author Contributions

**Conceptualization:** Dong Hyun Kim.

**Formal analysis:** Yun-Hee Choi.

**Methodology:** Yun-Hee Choi.

**Validation:** Martha Kim, Yoon-Hyeong Choi, Dong Hyun Kim.

**Writing – original draft:** Yun-Hee Choi.

**Writing – review & editing:** Martha Kim, Yoon-Hyeong Choi, Dong Hyun Kim.

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
