## [Decision Letter · Decision Letter 0]

28 Aug 2024

PONE-D-24-25756Adult male-specific inverse association between dry eye disease and intraocular pressure: KNHANES 2010–2012PLOS ONE

Dear Dr. Kim,

Thank you for submitting your manuscript to PLOS ONE. After careful consideration, we feel that it has merit but does not fully meet PLOS ONE’s publication criteria as it currently stands. Therefore, we invite you to submit a revised version of the manuscript that addresses the points raised during the review process. Please submit your revised manuscript by Oct 12 2024 11:59PM. If you will need more time than this to complete your revisions, please reply to this message or contact the journal office at plosone@plos.org. Please include the following items when submitting your revised manuscript:A rebuttal letter that responds to each point raised by the academic editor and reviewer(s). You should upload this letter as a separate file labeled 'Response to Reviewers'.A marked-up copy of your manuscript that highlights changes made to the original version. You should upload this as a separate file labeled 'Revised Manuscript with Track Changes'.An unmarked version of your revised paper without tracked changes. You should upload this as a separate file labeled 'Manuscript'.If applicable, we recommend that you deposit your laboratory protocols in protocols.io to enhance the reproducibility of your results. Protocols.io assigns your protocol its own identifier (DOI) so that it can be cited independently in the future. For instructions see: https://journals.plos.org/plosone/s/submission-guidelines#loc-laboratory-protocols. Additionally, PLOS ONE offers an option for publishing peer-reviewed Lab Protocol articles, which describe protocols hosted on protocols.io. Read more information on sharing protocols at https://plos.org/protocols?utm_medium=editorial-email&utm_source=authorletters&utm_campaign=protocols.

We look forward to receiving your revised manuscript.

Kind regards,

Daniel Duck-Jin Hwang

Academic Editor

PLOS ONE

Journal Requirements:

1. When submitting your revision, we need you to address these additional requirements. Please ensure that your manuscript meets PLOS ONE's style requirements, including those for file naming. The PLOS ONE style templates can be found at https://journals.plos.org/plosone/s/file?id=wjVg/PLOSOne_formatting_sample_main_body.pdf and https://journals.plos.org/plosone/s/file?id=ba62/PLOSOne_formatting_sample_title_authors_affiliations.pdf 2. Your ethics statement should only appear in the Methods section of your manuscript. If your ethics statement is written in any section besides the Methods, please move it to the Methods section and delete it from any other section. Please ensure that your ethics statement is included in your manuscript, as the ethics statement entered into the online submission form will not be published alongside your manuscript.

Additional Editor Comments:

It interests me that your study demonstrates a significant negative association between DED and IOP in Korean males using a large-scale general population epidemiological study. However, overall, there are several areas for improvement. For improvement, in addition to what I mentioned, please respond well to the points discussed below by two reviewers.

[Major comments]

1.In Table 1, it is noticeable that the percentage of DED cases among all subjects is 7.8% overall and 8.6% in urban areas, which appears to be significantly lower compared to the prevalence of DED in Asian countries in DEWSII epidemiology report. This is likely due to the criteria requiring both DED symptoms and a doctor's diagnosis. However, this may result in distortions due to patients' tendencies regarding hospital visits (i.e., whether they visit a hospital with the same degree of DED). This should be included as a limitation in the discussion.

2.In Table 2, there is no relationship between IOP and DED in Models 1 and 2, but there is a significant relationship in Model 3. The difference between Models 2 and 3 lies in the inclusion of family history of glaucoma, diabetes, and hypertension. It would be beneficial to add a detailed analysis of the significance of these differences between the models in the discussion. Additionally, it would be helpful to discuss the necessity of such a complex analysis system and explain the criteria used to classify risk factors into Models 1 to 3.

3.Regarding the sudden change in the percentage of DED cases by survey year in Table 1 (7.1% in 2010, 6.3% in 2011, and a sharp increase to 10.3% in 2012), how can this be explained? Such a drastic change could raise questions about the reliability of the data used in this analysis, so a possible explanation or acknowledgment of limitations is needed. Considering that only 55 out of 19,599 patients were undergoing glaucoma treatment, could this be considered too few, given the general prevalence of glaucoma in Korea? The use of glaucoma medication could be a significant variable in the results. Is there a possibility of insincere responses?

[Minor comments]

4. Lines 74–76: Did they really report that the non-glaucoma group with DED had significantly lower IOP than the non-glaucoma group without DED? Check it out. (PMID: 32811862)

5. It needs to be clearly stated whether the results of the multiple regression are beta values or odds ratios.

6. Tables S5 and S7 show positive values, unlike the other tables. In the text, there is a mention of "lower IOP," yet the title is set as "high IOP." It is necessary to check whether the descriptions, titles, and figures in these tables are accurate and consistent with the other tables.

7. The criteria for high IOP are defined as ">21" in S2 and "≥18" in S3, but there are no criteria listed for Tables 2, S5, and S7. Although line 125 of the paper states that high IOP is defined as">21," why is the criterion of"≥18" used in Table S3? This is very confusing.

8. Figure 2: The survey year and region, which were included in Model 1, are omitted. Please add them.

9. Lines 251–252: Shouldn't "significantly lower" be corrected to "significantly stronger negative correlation"? The meaning could be interpreted entirely oppositely.

10. Line 295: The expression "in vitro mouse study" seems odd. Please check the expression.

11. Lines 337–339: Alcohol leads to a reduction in the production of aqueous humor, which lowers IOP, so this is the opposite mechanism of increasing IOP by Alcohol. A more appropriate reason should be found.

Reviewers' comments:

Reviewer's Responses to Questions

**Comments to the Author**

1. Is the manuscript technically sound, and do the data support the conclusions?

Reviewer #1: Yes

Reviewer #2: Yes

2. Has the statistical analysis been performed appropriately and rigorously? 

Reviewer #1: Yes

Reviewer #2: Yes

3. Have the authors made all data underlying the findings in their manuscript fully available?

Reviewer #1: Yes

Reviewer #2: Yes

4. Is the manuscript presented in an intelligible fashion and written in standard English?

Reviewer #1: Yes

Reviewer #2: Yes

5. Review Comments to the Author

Reviewer #1: The relevance of reviewed article beyond doubt due to the prevalence of advanced dry eye syndrome. The study of the relationship of dry eye syndrome with various general and ophthalmological parameters (in particular, IOP) is of great interest for attention.

The article is presented in an accessible form and in understandable language. The abstract fully reflects the material presented. The tables are clear. The conclusions correspond to the stated goals.

The statistical methods that were used at each stage of the population study are sufficiently described. This proves the significance of the data analysis results.

During the review of the results, there were some questions that required clarification:

In the “Definition of DED” section of the “Materials and Methods” chapter, it would be worthwhile to describe in more detail which questionnaire is used to determine the definition of DED.

The article found that the presence of diabetes is determined based on several factors (fasting blood glucose levels, current diabetes medications or insulin injections, physician diagnosis). A factor such as fasting blood glucose level was measured separately or only depending on any of the other factors recognized by the authors (for example, fasting blood glucose level + doctor's diagnosis, etc.), or how an independent factor? If this factor was measured as independent did individuals with high fasting glucose levels have this individuals been additional measured glycated hemoglobin or did they have a glucose tolerance test performed?

Reviewer #2: This is an interesting paper that suggests possible relationship between dry eye disease and IOP by analyzing data of large population. I would like to recommend some revision before further consideration.

1. Introduction: Lines 55-79: description regarding IOP seems too long and somewhat distracting. Please make the description more compact.

2. Line 82 : Is there any reason the authors selected the data from KNHANES 2010-2012, as more recent data is available. Please describe the reason in the Discussion.

3. Discussion : sensation of dry eye symptoms can be association with increased systemic inflammatory level and pain modulation in the CNS. Would there be any possible relationship between the inflammation and IOP ? The authors can add some comments/

4. IOP measured with GAT may be affected by corneal hysteresis. Wouldn't there be some relationship between the corneal hysteresis and corneal sensation?

6. PLOS authors have the option to publish the peer review history of their article (what does this mean?). If published, this will include your full peer review and any attached files.

Reviewer #1: No

Reviewer #2: No

---

## [Author Response · Author response to Decision Letter 0]

21 Sep 2024

Response to Comments (Manuscript ID PONE-D-24-25756)

Adult male-specific inverse association between dry eye disease and intraocular pressure: KNHANES 2010–2012

Yun-Hee Choi, Martha Kim, Yoon-Hyeong Choi*, Dong Hyun Kim*

Journal Requirements:

Comment 1 

Response 1 We have confirmed that our manuscript meets PLOS ONS’s style requirements. Please see the revised manuscript.

Comment 2 

Response 2 We have revised that ethics statement should only appear in the Methods section. Please see the revised manuscript.

Comment 3 

Response 3 We have revised all our reference list complete and correct. There are no cited papers that have been retracted. Also, any changes to the reference list are described in the revision letter.

Additional Editor Comments:

General Comment 

It interests me that your study demonstrates a significant negative association between DED and IOP in Korean males using a large-scale general population epidemiological study. However, overall, there are several areas for improvement. For improvement, in addition to what I mentioned, please respond well to the points discussed below by two reviewers.

Response We deeply appreciate the constructive comments provided by the editor. We appreciate the editor’s constructive, detailed comments which helped us further improve our manuscript. Below we address your comments and list of changes that we made to our manuscript according to your reports. We believe that these modifications have strengthened the manuscript. We hope that the revised manuscript is suitable for publication in the PLOS ONE.

Comment 1 

[Major comments]

1.In Table 1, it is noticeable that the percentage of DED cases among all subjects is 7.8% overall and 8.6% in urban areas, which appears to be significantly lower compared to the prevalence of DED in Asian countries in DEWSII epidemiology report. This is likely due to the criteria requiring both DED symptoms and a doctor's diagnosis. However, this may result in distortions due to patients' tendencies regarding hospital visits (i.e., whether they visit a hospital with the same degree of DED). This should be included as a limitation in the discussion.

Response 1 We thank the editor for pointing out where additional clarity is needed. The reviewer is right. As we used both DED symptoms and doctor’s diagnosis to define DED, patient’s tendencies regarding hospital visits can result distortions. Based on the reviewer’s advice, we have revised the manuscript. Please see lines 368–370 in the revised manuscript.

Line 368: “Moreover, patients’ willingness to seek medical attention, despite experiencing similar symptoms, may lead to misclassification of DED.”

Comment 2 

2.In Table 2, there is no relationship between IOP and DED in Models 1 and 2, but there is a significant relationship in Model 3. The difference between Models 2 and 3 lies in the inclusion of family history of glaucoma, diabetes, and hypertension. It would be beneficial to add a detailed analysis of the significance of these differences between the models in the discussion. Additionally, it would be helpful to discuss the necessity of such a complex analysis system and explain the criteria used to classify risk factors into Models 1 to 3.

Response 2 We thank the editor for pointing out where additional clarity is needed. Based on the editor’s advice, we have added an explanation that, to examine the impact of different types of covariates on the results, we conducted analyses by dividing the covariates into following three distinct groups: 1) demographic variables, 2) anthropometric measurements and health-related behaviors, and 3) genetic or disease variables, to assess their influence on the outcome. Please see lines 163–166 and in the revised manuscript.

Line 163: “We developed three sequential models, differentiated by (1) demographic variables, (2) anthropometric measurements and health-related behaviors, and (3) genetic or disease variables, to assess their influence on the outcome [34].”

Reference: “Choi, Y. H., Huh, D. A., & Moon, K. W. (2021). Joint Effect of Alcohol drinking and environmental cadmium exposure on hypertension in Korean adults: analysis of data from the Korea National Health and Nutrition Examination Survey, 2008 to 2013. Alcoholism: Clinical and Experimental Research, 45(3), 548-560.”

The following references were additionally added. 

Choi, Y. H., Huh, D. A., & Moon, K. W. (2021). Joint Effect of Alcohol drinking and environmental cadmium exposure on hypertension in Korean adults: analysis of data from the Korea National Health and Nutrition Examination Survey, 2008 to 2013. Alcoholism: Clinical and Experimental Research, 45(3), 548-560.

Moreover, we have added a discussion in the interpretation section addressing why the inverse association between DED and IOP did not show significant effects in models 1 and 2, but emerged in model 3. Please see lines 341–349 in the revised manuscript.

Line 341: “As evidenced by multiple covariate adjustment models and subgroup analyses, the effect of DED on IOP varied according to demographic and disease-related characteristics. The inverse association between DED and IOP was not significant in models 1 and 2 (table 2), which adjusted for demographic characteristics and health-related behaviors. However, a significant association was observed in model 3, which additionally adjusted for genetic and disease-related factors, suggesting that family history of glaucoma, diabetes, and hypertension play crucial roles in this relationship. This finding was further elucidated in the subgroup analyses, which revealed a more pronounced inverse association between DED and IOP among males with higher household income and education levels who did not have hypertension.”

Comment 3 

3.Regarding the sudden change in the percentage of DED cases by survey year in Table 1 (7.1% in 2010, 6.3% in 2011, and a sharp increase to 10.3% in 2012), how can this be explained? Such a drastic change could raise questions about the reliability of the data used in this analysis, so a possible explanation or acknowledgment of limitations is needed. Considering that only 55 out of 19,599 patients were undergoing glaucoma treatment, could this be considered too few, given the general prevalence of glaucoma in Korea? The use of glaucoma medication could be a significant variable in the results. Is there a possibility of insincere responses?

Response 3 We thank the editor for the opportunity to further clarify these points. Although our study, which excluded some participants, observed an increase in DED prevalence from 6.3–7.1% in 2010–2011 to 10.3% in 2012, these findings closely align with another KNHANES 2010–2011 and 2012 study that reported physician-diagnosed DED prevalence of 8% and 10.4%, respectively, among the general adult population [Ahn et al., 2014; Kim et al., 2016]. Therefore, while our observed DED prevalence may be lower than typical estimates due to our use of both diagnostic and symptomatic criteria, it can be considered representative of the DED prevalence observed in Korean adults. 

Meanwhile, the increasing prevalence of DED over this period is primarily attributed to the rapid proliferation and use of smartphones and the rise in air pollution during this time period. We have incorporated these explanations into our discussion and limitations sections. Please see lines 277–281 and 366–368 in the revised manuscript.

Line 277: “This study utilized data from the KNHNAES (2010–2012), which represents the general population of South Korea, identified that the prevalence of DED increased from 7.1% to 10.3% during this period. This figure is similar to the 8% to 10.4% prevalence reported in national statistics for the same period [36,37], which is likely influenced by air pollution levels and the widespread use of smartphones [36,37].”

Line 366: “Second, DED was ascertained based on both self-reported symptoms and physician diagnoses. Therefore, DED prevalence reported in this study may differ from the actual prevalence in the Korean population [36,37].”

Reference: “Ahn, J. M., Lee, S. H., Rim, T. H. T., Park, R. J., Yang, H. S., Im Kim, T., ... & Society, O. (2014). Prevalence of and risk factors associated with dry eye: the Korea National Health and Nutrition Examination Survey 2010–2011. American journal of ophthalmology, 158(6), 1205-1214.

Kim, M., Oh, J. H., Park, C. Y., & Lee, S. W. (2016). Dry eye disease and allergic conditions: a Korean nationwide population-based study. American journal of rhinology & allergy, 30(6), 397-401.”

The following references were additionally added. 

Ahn, J. M., Lee, S. H., Rim, T. H. T., Park, R. J., Yang, H. S., Im Kim, T., ... & Society, O. (2014). Prevalence of and risk factors associated with dry eye: the Korea National Health and Nutrition Examination Survey 2010–2011. American journal of ophthalmology, 158(6), 1205-1214.

Kim, M., Oh, J. H., Park, C. Y., & Lee, S. W. (2016). Dry eye disease and allergic conditions: a Korean nationwide population-based study. American journal of rhinology & allergy, 30(6), 397-401.

Furthermore, it is important to note that the 55 individuals excluded due to ongoing glaucoma treatment do not represent the total number of patients undergoing glaucoma treatment in the general population. This subset was identified after sequentially excluding those without DED and IOP measurements (2,880 individuals) and those who had undergone cataract or refractive surgery (1,621 individuals) from the initial 19,599 participants. In accordance with the editor's suggestion and our own assessment that glaucoma medication could significantly impact the results, we chose to exclude these subjects from our analysis.

Lastly, given that the KNHANES is conducted with researchers explaining the questionnaire to participants, the likelihood of insincere responses is minimal. Even if such responses exist, they are likely too few to substantially influence the results.

Comment 4 

[Minor comments]

4. Lines 74–76: Did they really report that the non-glaucoma group with DED had significantly lower IOP than the non-glaucoma group without DED? Check it out. (PMID: 32811862)

Response 4 We thank the editor for the opportunity to further clarify this point. The authors report that the non-glaucoma group with DED had lower IOP than the non-glaucoma group without DED during spring to fall; however, they did not conduct statistical analysis to assess those relationships. As our previous descriptions were unclear, we have revised the sentences to better highlight statistically significant relationships between IOP and DED-related indicators. Please see lines 64–68 in the revised manuscript.

Line 64: “One study found a significant inverse correlation between IOP and corneal staining score, which showed higher values in DED patients, regardless of their glaucoma status [22]. Another study showed that IOP was significantly lower in patients with a short tear break-up time (TBUT), a characteristic of DED [23].”

Comment 5 

5. It needs to be clearly stated whether the results of the multiple regression are beta values or odds ratios.

Response 5 We thank the editor for the opportunity to further clarify these points. The results of multiple regression are beta values. Based on the reviewer’s advice, we have further clarified this point in the statistical analyses section. Please see lines 157–159 in the revised manuscript.

Line 157: “DED was categorized as a binary variable (presence or absence), while IOP was treated as a continuous variable. The association was assessed by comparing the effects of DED on IOP level and estimating regression coefficients (β) and 95% confidence intervals (CIs).”

Moreover, we have clarified the that the results of multiple linear regression analysis were β (95% CI) in Table 2. Please see the table 2 in the revised manuscript.

Table 2: “Multiple linear regression analysis results for the effects of dry eye disease on intraocular pressure (n = 13,194).”

Variables Total Male Female

 β (95% CI) β (95% CI) β (95% CI)

Model 1 

DED vs. no DED -0.005 (-0.021, 0.011) -0.023 (-0.051, 0.005) 0.001 (-0.019, 0.021)

Model 2 

DED vs. no DED -0.003 (-0.019, 0.0124) -0.024 (-0.052, 0.005) 0.003 (-0.017, 0.023)

Model 3 

DED vs. no DED -0.032 (-0.059, -0.004) -0.059 (-0.106, -0.012) -0.020 (-0.055, 0.015)

Model 1 

Bold: p < 0.05, Italic: p < 0.1

Model 1: adjustment for age, sex, survey year, region, income, and education

Model 2: model 1 + adjustment for alcohol drinking status, smoking status, exercise status, sleep duration, and body mass index

Model 3: model 2 + adjustment for family history of glaucoma, diabetes, and hypertension

Abbreviations: CI, confidence interval; DED, dry eye disease

Comment 6 

6. Tables S5 and S7 show positive values, unlike the other tables. In the text, there is a mention of "lower IOP," yet the title is set as "high IOP." It is necessary to check whether the descriptions, titles, and figures in these tables are accurate and consistent with the other tables.

Response 6 We thank the editor for the opportunity to check the descriptions of Table S5 and S7. The results presented in Tables S5 and S7 are derived from multiple logistic regression analyses, with odds ratios (ORs) as the primary measure. These ORs should be interpreted as indicating a negative association when less than 1 and a positive association when greater than 1. Consequently, the significant values shown in Model 3 of Tables S5 and S7 suggest that individuals with DED have a lower likelihood of experiencing high IOP (IOP) (>21 mmHg) compared to those without DED. We have revised the wording to avoid potential confusion arising from the term "Lower IOP." Additionally, Tables S5 and S7 have been renumbered to S4 and S6, respectively. Please see lines 268–272 in the revised manuscript.

Line 268: “According to the sensitivity analysis results of accounting for the left eye, DED and IOP showed inverse associations in the overall population along with males in the fully adjusted model (model 3; S1-S2 Tables). Even in models without complex sample weights, males with DED showed a significant inverse association with IOP in the fully adjusted model (model 3; S3-S4 Tables).”

Comment 7 

7. The criteria for high IOP are defined as ">21" in S2 and "≥18" in S3, but there are no criteria listed for Tables 2, S5, and S7. Although line 125 of the paper states that high IOP is defined as">21," why is the criterion of"≥18" used in Table S3? This is very confusing.

Response 7 We thank the editor for the opportunity to correct this point. Initially, we aimed to verify whether our results would hold using an alternative definition of high intraocular pressure (IOP) at ≥18 mmHg. However, we failed to adequately explain this in t

---

## [Decision Letter · Decision Letter 1]

20 Nov 2024

Adult male-specific inverse association between dry eye disease and intraocular pressure: KNHANES 2010–2012

PONE-D-24-25756R1

Dear Dr. Kim,

We’re pleased to inform you that your manuscript has been judged scientifically suitable for publication and will be formally accepted for publication once it meets all outstanding technical requirements.

Kind regards,

Daniel Duck-Jin Hwang

Academic Editor

PLOS ONE

Additional Editor Comments:

The authors have revised and supplemented their manuscript appropriately. Thank you for the diligent revisions made to the manuscript in line with our comments. I truly appreciate your hard work and dedication to improving the manuscript.

Reviewers' comments:

Reviewer's Responses to Questions

**Comments to the Author**

1. If the authors have adequately addressed your comments raised in a previous round of review and you feel that this manuscript is now acceptable for publication, you may indicate that here to bypass the “Comments to the Author” section, enter your conflict of interest statement in the “Confidential to Editor” section, and submit your "Accept" recommendation.

Reviewer #1: (No Response)

Reviewer #2: All comments have been addressed

2. Is the manuscript technically sound, and do the data support the conclusions?

Reviewer #1: Yes

Reviewer #2: Yes

3. Has the statistical analysis been performed appropriately and rigorously? 

Reviewer #1: Yes

Reviewer #2: Yes

4. Have the authors made all data underlying the findings in their manuscript fully available?

Reviewer #1: Yes

Reviewer #2: Yes

5. Is the manuscript presented in an intelligible fashion and written in standard English?

Reviewer #1: Yes

Reviewer #2: Yes

6. Review Comments to the Author

Reviewer #1: (No Response)

Reviewer #2: All the issues raised were adequately addressed. I believe now it can be published. Thank you for your effort.

7. PLOS authors have the option to publish the peer review history of their article (what does this mean?). If published, this will include your full peer review and any attached files.

Reviewer #1: No

Reviewer #2: No

---

## [Editor Report · Acceptance letter]

7 Jan 2025

PONE-D-24-25756R1 

PLOS ONE

Dear Dr. Kim, 

I'm pleased to inform you that your manuscript has been deemed suitable for publication in PLOS ONE. Congratulations! Your manuscript is now being handed over to our production team.

Kind regards, 

on behalf of

Professor Daniel Duck-Jin Hwang 

Academic Editor

PLOS ONE